# Impact of Medical Debt on the Financial Welfare of Middle- and Low-Income Families across China

**DOI:** 10.3390/ijerph17124597

**Published:** 2020-06-26

**Authors:** Jiajing Li, Chen Jiao, Stephen Nicholas, Jian Wang, Gong Chen, Jinghua Chang

**Affiliations:** 1Center for Health Economics Experiment and Public Policy, School of Public Health, Cheeloo College of Medicine, Shandong University, No. 44 Wenhua West Road, Lixia District, Jinan 250012, China; sdu_hn@163.com (J.L.); jiaochen987@163.com (C.J.); 2School of Economics and School of Management, Tianjin Normal University, No. 339 Binshui West Avenue, Tianjin 300387, China; stephen.nicholas@newcastle.edu.au; 3Guangdong Institute for International Strategies, Guangdong University of Foreign Studies, 2 Baiyun North Avenue, Guangzhou, Guangdong 510420, China; 4Top Education Institute, 1 Central Avenue, Australian Technology Park, Eveleigh, Sydney, NSW 2015, Australia; 5Newcastle Business School, University of Newcastle, University Drive, Newcastle, NSW 2308, Australia; 6Dong Fureng Institute of Economics and Social Development, Wuhan University, No. 54 Dongsi Lishi Hutong, Dongcheng District, Beijing 100010, China; wangjian993@whu.edu.cn; 7Center for Health Economics and Management, Economics and Management School, Wuhan University, Luojia Hill, Wuhan 430072, China; 8Institute of Population Research, Peking University, No. 5 Yiheyuan Road, Haidian District, Beijing 100871, China; chengong@pku.edu.cn

**Keywords:** medical debt, poverty, catastrophic health expenditure, medical expenditure, medical financial burden

## Abstract

Background: Medical debt is a persistent global issue and a crucial and effective indicator of long-term family medical financial burden. This paper fills a research gap on the incidence and causes of medical debt in Chinese low- and middle-income households. Method: Data were obtained from the 2015 China Household Finance Survey, with medical debt measured as borrowings from families, friends and third parties. Tobit regression models were used to analyze the data. The concentration index was employed to measure the extent of socioeconomic inequality in medical debt incidence. Results: We found that 2.42% of middle-income families had medical debt, averaging US$6278.25, or 0.56 times average household yearly income and 3.92% of low-income families had medical debts averaging US$5419.88, which was equivalent to 2.49 times average household yearly income. The concentration index for low and middle-income families’ medical debt was significantly pro-poor. Medical debt impoverished about 10% of all non-poverty households and pushed poverty households deeper into poverty. While catastrophic health expenditure (CHE) was the single most important factor in medical debt, age, education, and health status of householder, hospitalization and types of medical insurance were also significant factors determining medical debt. Conclusions: Using a narrow definition of medical debt, the incidence of medical debt in Chinese low- and middle-income households was relatively low. But, once medical debt happened, it imposed a long-term financial burden on medical indebted families, tipping many low and middle-income households into poverty and imposing on households several years of debt repayments. Further studies need to use broader definitions of medical debt to better assess the long-term financial impact of medical debt on Chinese families. Policy makers need to modify China’s basic medical insurance schemes to manage out-of-pocket, medical debt and CHE and to take account of pre-existing medical debt.

## 1. Introduction

It is no secret that bad health and bad debts often coincide. A large and growing worldwide problem, medical debt imposes financial hardship on household budgets. In the United States, the Kaiser Family Foundation reported that roughly a quarter of non-elderly American adults had difficulties paying their medical bills [1]. In many cases, health care costs substantially exceed patients’ ability to pay for that care, even among families with adequate health insurance coverage [2]. A U.S. national survey found that about 29 million people had a recent or accrued medical debt [3], and such debts were a leading contributor to personal bankruptcy. Large medical bills impose an additional sizable burden on those who are already economically fragile, where financially stretched households and households that are uninsured or underinsured can be pushed into long-term personal debt, bankruptcy and poverty [4]. One survey found that 44% of American families with medical debt exhausted their savings to service outstanding medical bills, and many of them traded medical debt for other types of debt [3]. Medical debt can be more damaging than other types of consumer debt because medical bills are frequently incurred through illness or injury that limits one’s ability to work. The work loss-medical debt problem confronts millions of middle- and low-income American families, where family savings are depleted paying medical debts, leaving families vulnerable to unanticipated economic shocks [4]. In Canada, the financial strain of medical costs rose because of medical expenditures and the loss of labor-related income due to illness [5]. The 2003 UK Commonwealth Fund Biennial Health Insurance Survey revealed that 37% of British adults had difficulty paying their medical bills, or had accrued medical debt, or both [2] 

Among developing economies, medical expenses impose the greatest financial pressures on low-income families and people suffering prolonged poor health, especially where families have accumulated unpaid medical bills [6]. Medical debt hardship in Romania sprang from the social health burden that grew from 64.6% of the total health services costs in 1998 to 82.7% of costs in 2004 [7]. Due to funding inadequacies, India’s public health infrastructure suffers perennial shortages, impacting the rural and poorer population segments the most. As a consequence of these health infrastructure shortages, nearly 70% of Indian health spending was borne as household medical debt, frequently to private health care facilities, leaving many families impoverished [8]. A Vietnamese survey revealed that the poor generally delay treatment and paid more for each process of illness than the rich. Further, poor Vietnamese families were compelled to cut consumption of essential goods or to borrow to meet their medical costs [9]. As in other developing countries, Cambodian households used a combination of savings, disposing of consumables and assets, and borrowed money to manage medical debt [10].

The vicious circle linking health and poverty and poverty and health remains a challenge in China. More than one in three Chinese households experienced difficulties in affording health care, or go without health care, or are impoverished by the health care costs [11,12]. Avoiding health care because of medical debts contribute to poor individual and national health outcomes [10]. A recent study found that medical debts of end-of-life Chinese cancer patients forced 83.92% of urban and 91.1% rural households below the poverty line [13]. After housing, business, education and cars, the 2012 China Household Finance Survey (CHFS) revealed that medical expenses were the next most important factor causing household debt [14]. Of course, the relationship between health debts and poverty is a global, not just a Chinese, phenomenon, [15] where out-of-pocket (OOP) expenses for public and private health care services drive many families into poverty and aggravate the poverty level of those who are already poor. China launched an ambitious national health reform plan in 2009, reaffirming the government’s central role in financing essential health care services, which focused on expanding health insurance coverage, strengthening the primary care system, providing public health services, and conducting public hospital reform [16,17].

As part of the reform process, the Chinese government established basic medical insurance schemes, which aimed to protect all families and individuals, particularly poor households, from financially catastrophic unaffordable medical care and medical debt induced poverty [18] Medical insurance reform supplemented China’s poverty alleviation program, which saw absolute rural poverty decreasing from 770 million in 1978 to 16.6 million in 2018 [19]. China’s basic medical insurance schemes comprised the Urban Employee Basic Medical Insurance (UEBMI) designed for employed urban residents; the Urban Resident Basic Medical Insurance (URBMI) designed for urban residents without formal employment (unemployed and retired), children and students; the New Cooperative Medical Scheme (NCMS) designed for rural residents; and Critical Illness Insurance (CII) for those whose hospitalization costs exceed a certain threshold. Health insurance coverage increased dramatically over the first decade of the 21st century, from 15% in 2000 to 96% in 2011 [20]. To protect families from heavy medical financial burden and impoverishment from medical expenses, health reform in China broadened the coverage of the medical insurance schemes and increased reimbursements, reducing OOP expenses.

While the total OOP medical expenses increased rapidly from RMB657.11 (US$95.2) billion in 2009 to RMB1513.36 (US$213.5) billion in 2017, the share of the OOP medical expenditure in total health expenditure decreased from 37.46% in 2009 to less than 30% in 2015, and remained stable through 2017 [21]. However, OOP medical expenses as a proportion of medical expenditure varied greatly across provinces, ranging from 16.29% in Beijing to 36.46% in Liaoning province, and the average cost of a single hospital admission could be equivalent to average person’s annual income, and more than twice the average annual income of those in the lowest 20% of the income distribution. As a result, many Chinese families with basic medical insurance face health cost financial burdens. Did China’s medical insurance schemes protect families from medical debt and poverty?

Surprisingly, few studies have explicitly examined medical service costs and medical insurance in relation to medical debt and poverty. Various Chinese studies on family finance classify medical debt as part of household consumption-fueled borrowing [22] or household debt [15,23] ignoring the particularities of medical debt. Using a 2004 cross-sectional survey of poor families enrolled in Medical Financial Assistance Scheme in countryside Chongqing, Hao et al. found that hospitalization costs and medical costs related to chronic diseases were significantly correlated with medical debt, and that the Medical Financial Assistance Scheme was limited in reducing medical debt [15]. In a study from two poor Chinese counties, Jiang et al. found that the medical debt incidence was 60%-70% among low- and middle-income households and the debt was not decreased by the CII between 2013 and 2016 [24]. Using data on the poorest families in 29 provinces in China, Jiang found more than 60% of families in poverty suffered from medical debt, no matter whether from urban or rural areas [23]. 

There is a paucity of nationwide studies on medical debt of middle- and low-income Chinese families and medical debt induced poverty. More generally, there are few international studies on health insurance, medical debt and poverty. To address this gap in the Chinese medical debt-poverty literature, this paper explores the impact of medical debt on the financial welfare of middle- and low-income families across China. We analyzed the relationship between medical debt and catastrophic health expenditures (CHE), calculated the proportion of Chinese households in medical debt and assessed the impact of medical debt and CHE on household poverty. Our study contributes to the wider international literature on the financial impact of medical debt, and the role of health insurance on medical debt, providing lessons for other developing countries by drawing on China’s experience with medical debt, OOP expenses, CHE and economic well-being.

## 2. Materials and Methods

### 2.1. Materials

In a sample of every family member in 37,289 households across 28 provinces in China, the 2015 China Household Finance Survey contained information on medical debt, medical expenditures, OOP medical expenses and household characteristics, including household income, household assets and household expenditures [14]. CHFS was the first and the only nationally representative survey on household finance in China [14]. Low-income families were defined as families in the last third of per capita income distribution and middle-income families were in the middle third. 

### 2.2. Variables

There is no agreed definition of medical debt. Some American research, such as Batty et al., referred to cases where a patient did not pay a bill for an unspecified period, but one long enough that the provider sends the medical debt to a collection agency [25] Some research defined medical debt as unpaid medical bills [26,27,28]. Several studies measured medical debt by the response to the following type of survey question: “If you added up all medical bills, about how much would they amount to right now? Including unpaid balance(s), or medical bills that are outstanding.” [29]. Several studies have found these measures of outstanding medical bills to be highly reliable medical debt proxies [4,30]. Carlson et al. defined medical debt as ‘how much money they currently owed health care providers, credit cards, or other loan companies for medical expenses’ [31] while Altice et al. used the depletion of assets to measure medical debt [32]. Some studies give medical debt a narrower definition, such as ‘borrow money or going into debt because of disease, its treatment, or the lasting effects of that treatment’ [33]. Hao et al. defined debt as money owed to clinics or others [15], while Jiang et al. defined the long-term medical debt burden as borrowed money [24].

We take a narrow definition of medical debt, based on the response to the 2015 CHFS question: ‘How much in total did your household borrow to pay medical debts?’ Similar to other studies, the answer included bank and credit union loans and private loans from friends and family [15,33]. Since our definition of medical debt does not take into consideration the fall in family savings or selling assets to cover medical expenditures, it provides a lower bound estimate of the level and amount of medical debt and medical debt induced poverty. 

From the CHFS, our independent variables on each household comprised all types of medical insurance, hospitalization, whether suffering non-communicated diseases (NCD), per capita household income, food and non-food household expenditures, CHE, household assets, OOP medical expenditure, and the marriage, age, education, working and health status of the householder. Since children and students were only covered by URBMI, and families bringing up children are more likely to face CHE, this study analyzed the ability of family members of different age groups insured by basic medical insurance to protect against a long-term medical burden. Based on the 2015 CFHS question “How much was paid or reimbursed by your family’s medical insurances?”, OOP medical payments were calculated, which included all types of health-related medical expenses that were not reimbursed through any health insurance [29]. The 2015 CFHS also included a question on medical expenditure before any health insurance reimbursements: “What was the amount your family spent on healthcare last year, including medical treatment expenses?”, which allows us to calculate the co-payments and deductibles. 

We define CHE as OOP healthcare payments comprising more than 40% of non-food household consumption, which reduces non-health household expenditure below the level required for necessities [34,35]. Per capita household income was calculated as the total family income divided by their household size [34] Households in poverty were defined as those who’s when total household expenditure was smaller than household subsistence spending, where subsistence spending was defined as the share of food expenditure in total household expenditure was at the 50th percentile [36]. Our household specific poverty measure varies from poverty measures based on a fixed poverty line for all households. Impoverished households were those who’s defined as when household expenditure was higher than subsistence spending, but was lower than subsistence spending net of out-of-pocket health payments [36]. Impoverished households were households tipped into poverty by medical expenses. The concentration index (CI) was employed to measure the extent of socioeconomic inequality (per capita income) in medical debt incidence [37,38]. Per capita income, household expenditure and assets, OOP medical expenses and medical debt were converted to US dollars at the 2014 average exchange rate.

### 2.3. Methods

Medical debt cannot be negative, which means that our dependent variable was truncated. To address the limited dependent variable problem, and that the influencing factors of medical debt of middle-income and low-income households may differ, we selected the hierarchical Tobit model [39,40]. We investigated the error distributions of the models to determine the level of heteroscedasticity, which was not a problem. The CHFS data contained the weight of each household, and the Tobit model estimation used appropriate sampling weights to account for the complex sample design and selection. The level of significance was set at α = 0.05, with an alpha level of *p* < 0.05 considered statistically significant. Stata SE 15.1 (StataCorp LLC, College station, TX, USA) was used to analyze the data. 

## 3. Results 

Table 1 displays the indicators of medical financial burden and medical debt of middle- and low-income families. The medical debt incidence of middle-income families was 2.42% of all middle-income families and 3.92% of low-income families, but only 0.93% for high-income families. The average medical debt of middle-income families in medical debt was US$6278.25 and US$5419.88 among low-income families, with the average medical debt of middle-income households 15.8% higher than that of low-income households. The median medical debt of low-income households was roughly 2.5 years their net household income and about 6 months’ median yearly income for medium-income households. Moreover, the CI in Table 1 for low and middle-income families’ medical debt was significantly pro-poor. 

The geographical distribution of the incidence of medical debt in Figure 1 shows that the incidence of medical debt was higher in northwest provinces (Heilongjiang, Inner Mongolia, Gansu, Qinghai, Yunnan) and Henan Province, and lower in eastern coastal provinces (Beijing, Shandong, Jiangsu, Shanghai, Zhejiang, Fujian). The higher medical debt incidence in the poorer more economically backward provinces was consistent with the CI results in Table 1. 

The mean OOP medical payments of low-income households were US$886.78 and US$925.71 for middle-income households. We found that 51.27% of low-income families and 33.90% of middle-income families with OOP payment faced CHE, with the share of OOP medical expenses in non-food expenditures of low-income families higher than that of middle-income families. Therefore, it was not surprising that Table 1 reports that CHE incidence in low-income families (36.92%) was significantly higher than middle-income families (25.54%). For the low-income households with CHE, 62.75% were in poverty while 41.80% of middle-income households were in poverty. 

We found that 9470, or 38.08% of middle- and low-income households fell below the poverty line because their total household expenditure was smaller than their food subsistence spending. As shown in Table 2, 44.44% of households in poverty suffered from CHE and 3.5% suffered from medical debt, which was significantly higher than that for non-poverty households, where 23.11% experienced CHE and 2.95% suffered medical debt (*p* < 0.05). While non-poverty households had a higher amount of OOP medical expenditure and medical debt than households in poverty, these non-poverty households had a greater capacity to make OOP and medial debt payments.

We calculated that 1534 (or 9.96%) non-poverty household were impoverished by CHE, when these households were tipped into poverty after incurring OOP expenditures for health services. As Table 3 shows, among these impoverished families, the incidence of CHE reached 79.01%, and the incidence of medical debt was 11.08%, which was significantly higher than non-impoverished households (*p* < 0.001), where CHE incidence was only 16.76% and medical debt was 2.05%. Impoverished households and non-impoverished households had similar amounts of medical debt, but the former had much higher OOP medical expenditure.

Table 4 shows the Tobit regression model results for the influencing factors on medical debt of low- and middle-income families. Middle-income households’ medical debt was affected by per capita income and household assets, while low-income households were not. Whether CHE occurred, and the amount of OOP medical expenditure, significantly affected the medical debt for all families. CHE increased household medical debt by US$18.31 for middle-income households and US$16.26 for low-income households. Age and education level of family members and the health status of householder also had a significant impact on medical debt among low-income families. When the health situation of the householder deteriorated by one level of the five ordered categories, the medical debt of low-income families increased by US$11.07, and that of middle-income families increased by US$13.31. Compared to the influencing factors of medical debt in middle-income families, debt of low-income families was also impacted by the middle-aged members suffering from NCD, raising medical debt by US$14.08. Hospitalization was an important factor affecting family medical debt. Middle-aged members’ and students’ hospitalization significantly increased the medical debt of low-income families by US$9.28 and US$45.90 for middle-income families. The hospitalization of the elderly significantly increased the medical debt of middle-income families by US$14.60. We also tested for rural-urban differences, but this variable was not significant.

Since the basic medical insurance in China was not planned for children and students, the impact of their medical insurance and health service utilization on family’s medical debt was complex. Among all kinds of basic medical insurance, only the URBMI held by the elderly effectively reduced middle-income families’ medical debt by US$19.27. China’s existing medical insurance policy offered little medical debt protections for middle-aged people and children under 5 years, which was reflected in the result that the URBMI held by middle-age members increased the medical debt in middle-income households and children under 5 years old insured by the NCMS increased the medical debt of middle-income families. 

## 4. Discussion 

Although health care expenses impose a heavy financial burden on family budgets, there are few studies on household medical debt or the long-term financial burden of medical debt in China. Compared to other medical debt research, there are two characteristics of medical debt of Chinese households. The first is the low incidence. Our analysis found that the medical debt incidence of middle-income families was only 2.42% of all middle-income families and 3.92% for low-income families. Second, once medical debt occurs, medical debt brings serious financial and long-term negative impacts on families, representing 6-month household income for middle-income families and 2.5 years’ household income for low-income families.

In comparison with medical debt studies in America [1], Britain [2], India [8] and Cambodia [10], medical debt incidence of Chinese households was low. We took a narrow measure of medical debt, defined as debt incurred by outside borrowing for medical expenses [15,33]. Using household savings and disposal of assets to pay for medical expenses were not included, resulting in a lower bound estimate of the incidence of medical debt. With the largest per capita savings rate in the world, Chinese families’ frugal habits would have seen them dispose of their assets or use their accumulated savings to pay medical expenses before seeking external borrowed funds. We can get some measure of households selling assets to meet medical debt by comparing the household assets of debt and non-debt households. The average household assets of medical debt households were US$34157.60 compared to US$86998.18 for non-medical debt households, which suggests assets were sold before borrowing to pay off medical debt. These non-borrowing debt payment strategies should be categorized into a broader definition of medical debt. Since data on non-borrowing for medical debt strategies were not covered in existing CFHS surveys, new surveys need to be undertaken to measure these alternative strategies, especially using accumulated saving and assets dispersal for covering medical debt. Moreover, the Chinese government has a scheme of Medical Financial Assistance and CII, providing part medical financial support for impoverished households due to illness, which would reduce the medical debt of low-income families to a certain extent. These schemes should be promoted, expanded and made accessible to a large share of Chinese households. 

Not surprisingly, medical debt was one of the most common debts for middle- and low-income families, which is similar to finding from other Chinese and international studies. Although the overall incidence of medical debt in China is low, once medical debt happens, it had a long-term negative impact on the financial sustainable development of middle- and low-income families. This contrasts with an American study that found most medical debts were relatively modest in size [25] with more than 50% of them less than $600 annually [41] According to other US surveys, the average debt load per person was about half of their annual reported income [42,43]. However, low-income families in China required several years’ annual income to repay their medical debts, and even medium-income households medical debt accounted for roughly half their yearly income. Given the need to use part of annual household income for day-to-day living expenses, medical debts took many years to be repaid, indebting households for years as medical cost repayments were made. China’s experience might reflect the impact of medical debt on household finances of other developing countries. Further studies in China and in other countries should investigate medical debt as a powerful long-term financial burden on families and to expand the definition of medical debt from borrowings to broader definitions of medical debt.

Such high medical debt to family income ratios were likely to have irreversibly damaged Chinese household finances. Some families with medical debt were condemned to years of medical debt repayment, while other families were impoverished by their inability to pay their medical debts. This health-induced poverty is a global phenomenon, where medical debt was an important cause of family bankruptcy in the United States [4]. Although there is no system of family bankruptcy in China, the study of extreme poverty caused by medical debt requires attention, and was likely the experiences of families experiencing medical debt in other developing countries. 

CHE, which was often used to measure the medical financial burden based on family income, had a significant impact on medical debt and on family financial wellbeing. We found that 62.55% of households with medical debt suffered from CHE, with only 30.21% of households without medical debt suffering CHE. The greater the CHE’s affect, the higher the medical debt relative to household income for both middle- or low-income families. Similar to previous studies [44,45], we found that medical debt and CHE run in parallel. When CHE occurs, nearly 10% of meddle- and low-income non-poverty households were tipped into poverty and households in poverty sunk deeper into poverty. For households in poverty, further medical treatments will significantly increase the risk of incurring additional medical debt, reducing the likelihood that families will seek needed medical attention. Considering the possible long-term impact of CHE and medical debt on families, including poverty, reduced health access and poorer health status, health insurance policymakers should aim to structure insurance schemes to attenuate OOP expenses, reducing both CHE and medical debt. China’s need to reassess the structure of their insurance schemes to address OOP expenses, CHE and medical debt provides a general lesson for developing countries with emerging national health insurance programs. From 2015 to 2017, the proportion of OOP medical expenses to total medical expenses did not change significantly, so long-term medical debt and CHE is an ongoing problem for many Chinese families. 

In addition to CHE, we found that health status, working conditions, hospitalization and medical insurance of middle-aged people were crucial factors determining medical debt for middle- and low-income families. Under China’s existing insurance schemes, the OOP medical expenditure of hospitalization services posed a large financial burden for low-income families, which, for example, led to a US$45.90 promotion in household medical debt when it occurred to students. Similar to US empirical studies [4], we found that hospitalization was one of the leading factors to medical debt [15]. Our results suggest that future studies in China and other developing countries should consider the interplay of health status, working conditions, hospitalization and medical insurance on medical debt. 

Unfortunately, China’s basic health insurance schemes played a limited role in protecting against medical debt. We found that only middle-aged people insured by the URBMI effectively reduce their medical debt, while people insured by the NCMS or UEBMI increased their exposure to medical debt. The reimbursement policies of the NCMS and UEBMI were inadequate and need to be revised. In particular, URBMI reimbursement policies for children under 5 years old and NCMS reimbursement policies for children and students should be further revised to effectively improve health insurance protection for low-income households. Further, the basic medical insurance schemes do not take into account the family income and economic status of the insured, which forced low and middle-income families to borrow to pay-off high medical expenses. In spite of the 2009 health reforms, further health reform, including health insurance reform, remains.

We suggest that medical insurance and other income relief funds should identify existing medical debts when compensating patients for new medical expenses. By providing appropriate financial support to families with medical debts, access and equity in health will be improved and families further relieved from the economic burden of disease caused debt. Awareness of existing medical debt and household health-related impoverishment will also help to minimize the likelihood that policy reforms create unintended consequences to these vulnerable groups. Policymakers should define low-income households as a sub-population “at risk” of financial failure from CHE and medical debt and implement insurance scheme support. Further, “at risk” families require safety nets to increase access to healthcare without incurring additional medical debt [44]. The lessons from China can inform health policy makers in other developing countries, in designing insurance coverage, CHE insurance and safety nets for medical debt.

This study has a number of limitations. We used a narrow definition of medical debt, which did not include using savings and disposal of assets to pay-off medical expenditures, leading to a lower bound estimate of medical debt. Future studies in China and other developing countries with similar saving habits should consider the appropriation of savings when analyzing medical debt. We found households in western China were more likely to incur medical debt than other regions according to the Figure 1. However, the CHFS did not cover two western autonomous provinces, Tibet and Xinjiang, which should be examined in future studies. Since CHFS is a retrospective survey, the recall bias and reporting bias of debts are inevitable in the survey. We did not assess whether medical debt incidence was related to the provision of private health services in the region, which future studies should address. Finally, the database used cross-section data, and future longitudinal population-based studies are warranted to improve the understanding of the magnitude and extent of financial hardship among middle- and low-income families. 

## 5. Conclusions

This study fills a research gap on medical debt in China. Using a narrow definition of medical debt, the incidence of medical debt in China was relatively low, 2.42% of all middle-income families and 3.92% of low-income families. However, once medical debt occurred, it had a long-term negative impact on the financial sustainable development of families, tipping many low and middle-income households into poverty and imposing several years of debt repayment on households. We recommend further studies using broader definitions of medical debt and other developing countries with similar saving habits to China. Medical debt was a complex mix of household income, OOP expenses, CHE, type of medical insurance, health status, working conditions and hospitalization, which varied across low and middle-income households. Policy makers need to consider this mix of factors when addressing the medical debt problem. In particular, China’s basic medical insurance schemes needs to consider reforms to address vulnerable families based on household income levels, CHE and pre-existing medical debts.

## Figures and Tables

**Figure 1 ijerph-17-04597-f001:**
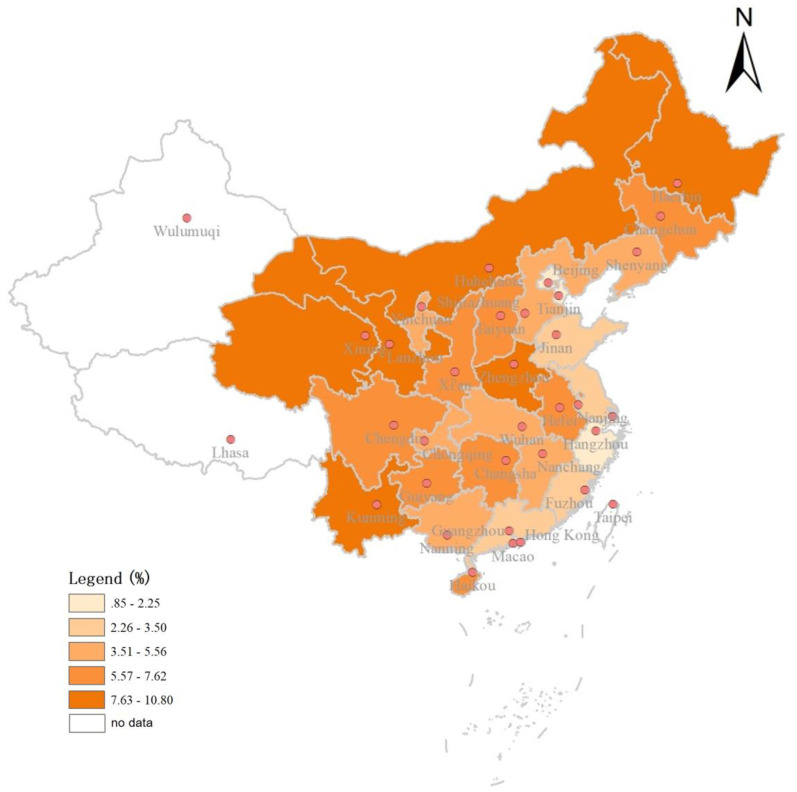
Geographical Distribution of Medical Debt Incidence in 2015.

**Table 1 ijerph-17-04597-t001:** Medical Financial Burden of Middle- and Low-Income Families.

Variable	Low-Income Households		Middle-Income Households	
	*n*	Median or Proportion	*n*	Median or Proportion
Median medical debt (US$)	487	3268.51	301	4085.63
Median per capita income (US$)	12435	738.03	12433	3757.88
Median years to pay medical debt (years)	487	2.49	301	0.56
CI for medical debt incidence	12435	−0.070 (*p* = 0.007)	12433	−0.306 (*p* = 0.000)
Median OOP payment (US$)	9409	244.19	9336	325.58
CHE incidence (%)	12435	36.92	12433	25.54
CHE forcing households into poverty (%)		62.75		41.80

**Table 2 ijerph-17-04597-t002:** Catastrophic Health Expenditure Incidence and Medical Debt among Poverty and Non-Poverty Households.

Variables	Households in Poverty		Non-Poverty Households		χ^2^
	n	Median or Proportion	n	Median or Proportion	
**CHE incidence**	9470	44.44%	15398	23.11%	*p* < 0.001
**The proportion of families with medical debt**	9470	3.50%	15398	2.95%	*p* = 0.017
**median medical debt (US$)**	331	1953.51	457	4069.81	
**median OOP payment (US$)**	7250	162.79	11525	325.58	

**Table 3 ijerph-17-04597-t003:** Medical Financial Burden among Impoverished and Non-Impoverished Households.

Variables	Impoverished Households		Non-Impoverished Households		χ^2^
	n	Median or Proportion	n	Median or Proportion	
CHE incidence	1534	79.01%	13864	16.76%	*p* < 0.001
The proportion of families with medical debt	1534	11.08%	13864	2.05%	*p* < 0.001
median medical debt (US$)	170	4069.81	284	4232.60	
median OOP payment (US$)	1534	1627.92	9991	325.58	

**Table 4 ijerph-17-04597-t004:** Tobit Regression Model of Influencing Factors on Medical Debt of Low- and Middle-Income Families.

Variable	Low-Income Households		Middle-Income Households	
	b	*p*	b	*p*
per capita income (RMB 10 thousand≈US$1645.25)	-	-	–2.391	0.000
OOP payment (RMB 10 thousand≈US$1645.25)	0.524	0.000	0.886	0.000
CHE	0.254	0.000	0.286	0.011
Household asset (RMB 10 thousand≈US$1645.25)	-	-	–0.054	0.000
householder age	–0.009	0.002	-	-
householder education	–0.101	0.000	-	-
householder working status	0.121	0.082	0.128	0.065
householder health status	0.173	0.000	0.208	0.000
middle-age members with NCD	0.220	0.000	-	-
middle-age members insured by NCMS	-	-	0.346	0.000
middle-age members uninsured	-	-	0.182	0.128
middle-age members hospitalized ever	0.145	0.050	0.115	0.127
children under 5 uninsured	–0.512	0.000	–0.221	0.134
children under 5 hospitalized ever	0.294	0.117	-	-
children under 5 insured by URBMI	-	-	0.454	0.019
students hospitalized ever	0.717	0.000	-	-
students insured by URBMI	–0.290	0.145	-	-
the elderly insured by URBMI	-	-	–0.301	0.037
the elderly hospitalized ever	0.200	0.066	0.228	0.050
constant	–1.606	0.000	–2.158	0.000
uncensored	483		301	
Log pseudo likelihood	–0.957		–0.309	

Notice: ‘-’ means this variable was not selected by stepwise selection in Tobit regression model.

## Data Availability

The data that support the findings of this study are available from the Survey and Research Center for China Household Finance, but restrictions apply to the availability of these data, which were used under license for the current study, and so are not publicly available. Data are however available from the authors upon reasonable request and with permission of the Survey and Research Center for China Household Finance on the website http://chfs.swufe.edu.cn/datas/.

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
