# Peer review of "Impact of Medical Debt on the Financial Welfare of Middle- and Low-Income Families across China"

_ijerph, 2020, doi:10.3390/ijerph17124597_

Round 1

Reviewer 1 Report

Manuscript ID: ijerph-792809

Title    Medical Debt and Catastrophic Health Expenditures among Middle- and Low-Income Families in China.

Brief summary.

This paper faces the issue of medical debt as effective indicator of long-term family medical financial burden.

The paper explores the impact of medical debt on the financial welfare of middle- and low-income families across China.

Data came from the 2015 China Household Finance Survey and medical debt was measured as borrowings from families, friends and third parties. Tobit regression models were used to analyze the data. Finally, the concentration index was employed to measure the extent of socioeconomic inequality in medical debt incidence. In my opinion the paper presents a good contribution both in terms of methods and of empirical evidence. The topic is interesting. Data and results supported the conclusions.

According to this, this reviewer wishes to express gratitude to the authors for the opportunity of reading this interesting and inspiring paper.

However, the study requires some revisions in order to improve the quality of the paper and prior to being published in International Journal of Environmental Research and Public Health.

Therefore, I invite authors to respond to my comments and revise their manuscript.
Here are my key observations on the content and quality of the manuscript.

Broad comments.

Title.

In my opinion the title could be more appropriate in the following form: “Impact (or effect) of medical debt on the financial welfare of middle- and low-income families across China”, or “Incidence and 20 causes of medical debt in Chinese low and middle-income households”. In one of these forms, the title allows the reader to understand rapidly the aim of the paper.

Method.

Please offer proper justification of why Tobit regression was applied in the manuscript. In particular, the paper lacks appropriate references which justify and allow to understand the method used. Also, could be useful to clearly describe the research questions and how these are being addressed.

Discussion.

Your manuscript lacks discussion on how other countries could benefit from this research. In particular, in describing a Case Study, it’s important from the authors to offer a detailed discussion and / or enunciation of (a) lessons learned in applying a certain technique/method, (b) how the experience of the case study could be replicated to another entity in other parts of the world to enhance its broader impact.

This reviewer wishes the authors good luck with the publication of their work!

Author Response

Dear Reviewer,

We have addressed the comments by each Reviewer, providing page and line numbers where we have addressed each comment.

We have also revised the paper for English expression and clarity of argument.

We wish to thank the Reviewer for their insightful comments. We have addressed each comment by Reviewer 1, indicating by page and line number our revision.

Title.

In my opinion the title could be more appropriate in the following form: “Impact (or effect) of medical debt on the financial welfare of middle- and low-income families across China”, or “Incidence and 20 causes of medical debt in Chinese low and middle-income households”. In one of these forms, the title allows the reader to understand rapidly the aim of the paper.

Thanks for your suggestion. We accept the first option ‘Impact of medical debt on the financial welfare of middle- and low-income families across China’ as the title.

Method.

Please offer proper justification of why Tobit regression was applied in the manuscript. In particular, the paper lacks appropriate references which justify and allow to understand the method used. Also, could be useful to clearly describe the research questions and how these are being addressed.

Thanks for your suggestion.

On page 4 lines 191-194 we have provided a justification for the use of the Tobit model with appropriate references.

On page 4, lines 144+ we clearly describe the research questions.

Discussion.

Your manuscript lacks discussion on how other countries could benefit from this research. In particular, in describing a Case Study, it’s important from the authors to offer a detailed discussion and / or enunciation of (a) lessons learned in applying a certain technique/method, (b) how the experience of the case study could be replicated to another entity in other parts of the world to enhance its broader impact.

We draw out lessons and experiences for other countries in several place in the Discussion.

Page 8 lines 277+, comparisons are made with 5 other countries. We suggest reasons why the incidence of medical debt was lower in China, including the savings rate and China’s insurance schemes to protect poor households (See page 8, lines 281-294)

Page 9 line 298+, we discuss the impact of medical debt on households in China versus the US, and suggest that China’s experience might reflect that of other developing countries (see page 9, lines 306-307). Page 9, line 307-309 of our Discussion, we declare that ‘Further studies in China and in other countries should investigate medical debt as a powerful long-term financial burden on families and to expand the definition of medical debt from borrowings to broader definitions of medical debt.

Page 9 lines 315-316, we make explicit the lessons for developing countries.

Page 9 lines 329-331 makes the point that other developing countries can learn from China’s need to revise its health schemes to address OOP, CHE and medical debt.

Page 9, lines 339-342 of discussion, we recommend that future studies in China and developing countries should consider the interplay of influencing factors in explaining medical debt.

Page 10, lines 366-367 and 371-372, we recommended that in China and countries with similar saving habits, future studies should consider the misappropriation of savings when analyzing their own medical debt and the connection between saving and long-term financial burden.

Reviewer 2 Report

Comments

  1. The contribution to the (international) literature should be stated in the revised introduction.
  2. Do the survey data contain weights? Why they have not been used in the estimations?
  3. Is medical debt incidence related to the provision of private health services in the region?
  4. The concluding section should state practical policy-related conclusions.

Author Response

Dear Reviewer,

We have addressed the comments by each Reviewer, providing page and line numbers where we have addressed each comment.

We have also revised the paper for English expression and clarity of argument.

We wish to thank the Reviewer for their insightful comments. We have addressed each comment by Reviewer 2, indicating by page and line number our revision.

  1. The contribution to the (international) literature should be stated in the revised introduction.

    On page 3, lines 125-134, we have revised the Introduction to reference the international literature.

  1. Do the survey data contain weights? Why they have not been used in the estimations?

    On page 5, line 195-197, we state that the CHFS data contained the weight of each household, and the Tobit model estimation used appropriate sampling weights to account for the complex sample design and selection.

  1. Is medical debt incidence related to the provision of private health services in the region?

    It is a good question. However, we did not address this issue. We have added this in the Limitations and future research section on page 10, lines 371-373.

  1. The concluding section should state practical policy-related conclusions.

    We have made practical policy implications on page 9, lines 340-342; page 9, lines 346-349; page 10, lines 353-363. Practical policy advice for other developing countries is provided on Page 9 line 298+; Page 9 lines 315-316; Page 9 lines 329-331; Page 9, lines 339-342.

Round 2

Reviewer 1 Report

My comments have been addressed by the authors.

Now the paper is well written, with clear structure and careful explanations, enabling others to replicate these techniques if desired. The quality of experimental data is convincing, and the conclusions appear to be reliable.

This reviewer wishes the authors good luck with the publication of their work!